

**Secondary Organic Aerosols Derived from Intermediate Volatility**
**n-Alkanes Adopt Low Viscous Phase State**
Tommaso Galeazzo[1], Bernard Aumont[2], Marie Camredon[2], Richard Valorso[2], Yong B. Lim[3],
Paul J. Ziemann[4,5], and Manabu Shiraiwa[1,*]
1. Department of Chemistry, University of California, Irvine, CA92625, USA
2. Univ Paris Est Creteil and Université Paris Cité, CNRS, LISA, F-94010 Créteil, France
3. California Air Resources Board, Riverside, CA92507, USA
4. Department of Chemistry, University of Colorado, Boulder, Colorado, USA
5. Cooperative Institute for Research in Environmental Sciences (CIRES), University of
Colorado, Boulder, Colorado, USA
* Correspondence to: m.shiraiwa@uci.edu



**Abstract.**
Secondary organic aerosol (SOA) derived from n-alkanes, as emitted from vehicles and volatile
chemical products, is a dominant component of anthropogenic particulate matter, yet its
chemical composition and phase state are poorly understood and hardly constrained in aerosol
models. Here we provide a comprehensive analysis of n-alkane SOA by explicit chemistry
modeling, machine learning, and laboratory experiments to show that, counterintuitively, n-
alkane SOA adopt low viscous semisolid or liquid states. Our study underlines the complex
interplay of molecular composition and SOA viscosity: n-alkane SOA with higher carbon
number mostly consists of less functionalized first-generation products with lower viscosity,
while the lower carbon number SOA contains more functionalized multigeneration products
with higher viscosity. This study opens up a new avenue for analysis of SOA processes and the
results indicate little kinetic limitations of mass accommodation in SOA formation, supporting
the application of equilibrium partitioning for simulating n-alkane SOA formation in large-
scale atmospheric models.

**Introduction**
Secondary organic aerosols (SOA) are ubiquitous in the atmosphere, affecting climate, air
quality and public health (Pöschl and Shiraiwa, 2015; Jimenez et al., 2009). They are generally
formed by multigenerational oxidation of volatile organic compounds (VOCs) emitted by both
anthropogenic and biogenic sources followed by condensation of semi-volatile oxidation
products into the particle phase (Ziemann and Atkinson, 2012; Kroll and Seinfeld, 2008). As
an important class of SOA precursors, there is a growing attention to intermediate volatile
organic compounds (IVOCs), which can partition to the gas phase upon dilution of primary
organic aerosols after fresh emission sources such as vehicle tailpipes, combustion of fossil and
fuel oils, and volatile chemical products (Robinson et al., 2007; Mcdonald et al., 2018). The
inclusion of IVOCs in the model simulations helps to reduce the gap between model simulation
and field observation of SOA (De Gouw et al., 2011; Li et al., 2022; Zhao et al., 2016).

SOA can adopt different particle phase states (liquid, amorphous semisolid, and glassy

solid), depending on its chemical composition, relative humidity and temperature (Virtanen et
al., 2010; Petters et al., 2019; Reid et al., 2018; Renbaum-Wolff et al., 2013) and also evolving
upon chemical aging and photochemistry (Baboomian et al., 2022). SOA phase state plays an
important role in a number of atmospheric multiphase processes (Shiraiwa et al., 2017). The
occurrence of glassy SOA in the free troposphere can impact activation pathways of ice crystals
and cloud droplets (Knopf and Alpert, 2023). Slow diffusion in viscous particles induces kinetic



limitations in heterogeneous and multiphase reactions (Zhang et al., 2018; Zhou et al., 2019;
Shiraiwa et al., 2011), affecting long-range transport (Shrivastava et al., 2017; Mu et al., 2018).
The timescale of SOA partitioning can be prolonged in viscous particles (Schervish and
Shiraiwa, 2023), retarding uptake of semi-volatile compounds and mixing of different particle
populations (Ye et al., 2016). Particle phase state also modulates SOA growth to cloud
condensation nuclei sizes, affecting cloud life cycle (Zaveri et al., 2022). While the phase states
of SOA generated by biogenic VOCs such as terpenes and isoprene have been extensively
studied (Virtanen et al., 2010; Petters et al., 2019; Renbaum-Wolff et al., 2013; Baboomian et
al., 2022; Zhang et al., 2018; Zhou et al., 2019), those derived from IVOCs are hardly
investigated and remain poorly constrained.

Viscosity ($\eta$) is a dynamic property that characterizes the particle phase state, which can

be derived from the glass transition temperature ($T_g$) of the constituting species (Koop et al.,
2011; Shiraiwa et al., 2011). Several structure-activity relationships models have been
developed to predict the $T_g$ of an organic compound using various molecular properties
including molar mass, atomic O:C ratio (Shiraiwa et al., 2017), elemental composition (Derieux
et al., 2018), and volatility (Li et al., 2020; Zhang et al., 2019). These parameterizations do not
consider molecular structure nor functionality explicitly. Galeazzo and Shiraiwa (2022)
overcame this limitation by developing a machine learning-based model, tgBoost, with an
application of cheminformatics "molecular embeddings" that retains detailed information on
atomic composition, molecular structure and connectivity (Galeazzo and Shiraiwa, 2022). The
main novel feature introduced by tgBoost is model capability to predict different $T_g$ for
structural isomers and high sensitivity of $T_g$ to various functional groups, consistent with
viscosity measurements for functionalized compounds (Rothfuss and Petters, 2017; Grayson et
al., 2017).

Long-chain linear alkanes (n-alkanes) are representative IVOCs and account for a

substantial fraction of non-methane hydrocarbons in urban air as mainly emitted from
anthropogenic activities such as vehicle exhausts and incomplete fuel combustion (Li et al.,
2022). Gas-phase oxidation of n-alkanes by OH radicals can trigger the formation of SOA with
high yields, as observed in laboratory experiments (Aimanant and Ziemann, 2013a; Lim and
Ziemann, 2009b) and field observations (Gentner et al., 2012; Li et al., 2022). Gas-phase
oxidation pathways of n-alkanes are relatively well understood and successfully simulated by
detailed gas-phase chemistry modeling (Aumont et al., 2012; La et al., 2016), but the chemical
composition of n-alkane SOA has only been fully characterized for the $C_{16}$ n-alkane (Ranney
et al., 2023) and the phase state and viscosity of alkane SOA are unknown. Therefore, the n-



alkane SOA system provides an ideal benchmark for the investigation of the interplay of
chemical composition, particle phase state and kinetic limitations influencing SOA growth and
evolution.

In this study, we implemented tgBoost in an explicit gas-phase chemistry model

GECKO-A to investigate the complex interplay of chemical composition, kinetic partitioning,
and phase state of n-alkane SOA generated under dry and high NOx conditions. The GECKO-
A model is to date one of the most comprehensive generators of gas-phase chemical schemes,
as it automatically generates detailed gas-phase chemical mechanisms involving thousands to
millions of oxidation products from a given VOC precursor based on established reaction
pathways and structure–activity relationships (Aumont et al., 2012; La et al., 2016). The
simulations were conducted with variable effective mass accommodation coefficient to
consider potential kinetic limitations in amorphous semisolid particles (Shiraiwa and Pöschl,
2021). The simulated results were compared with chamber experimental data on SOA yields
(Lim and Ziemann, 2009b) as well as new measurements on thermal desorption temperatures
and functional group distributions.

**Methods:**
**Model simulations.**

We applied the Generator for Explicit Chemistry and Kinetics of the Organics in the

Atmosphere (GECKO-A) (Aumont et al., 2012; La et al., 2016) to obtain detailed reaction
schemes of gas-phase OH oxidation of n-alkanes along with rate constants. The GECKO-A
generator used for oxidation of linear n-alkanes includes the latest structure-activity
relationships to treat the chemistry of organic compounds with OH radical (Jenkin et al., 2018a,
b; Jenkin et al., 2019), the bimolecular reactions of peroxy radicals (Jenkin et al., 2019), as well
as alkoxy radical decomposition and H migration reaction rates (Vereecken and Peeters, 2009;
La et al., 2016). The vapor pressures of semi-volatile species were estimated by using
Nannoolal's group contribution method (Nannoolal et al., 2008) implemented in GECKO-A,
as described in detail in Valorso et al. (2011). Species with vapor pressure below $10^{-13}$ atm are
assumed to be of low enough volatility to completely partition to the condensed phase and their
gas phase chemistry is then not generated in the mechanism. The model treats unimolecular
particle-phase reactions including cyclization of hydroxyketones and dehydration of cyclic
hemiacetals to form dihydrofurans (La et al., 2016). The model does not treat autoxidation and
dimerization in the gas phase, but these processes should be minor pathways during n-alkane
oxidation in the presence of high NOx as the reaction of peroxy radicals with NOx should be



dominant (Praske et al., 2018; Pye et al., 2019); thus, their absence from GECKO-A chemical
schemes should not have major impacts on the simulated results.
These explicit chemical mechanisms were implemented into a box model to simulate
the multigenerational oxidation of n-alkanes, partitioning of oxidation products into the particle
phase based on their vapor pressures, and vapor wall loss to mimic chamber experiments (La
et al., 2016). We replicated the experimental conditions used in Lim and Ziemann (2009b) to
generate SOA from OH oxidation of $C_8$-$C_{17}$ n-alkanes at high $NO_x$ conditions in the presence
of non-volatile dioctyl sebacate (DOS) seed particles with particle radius of 150 nm and mass
loading of 200 $\mu g\ m^{-3}$. Temperature was held constant at 295.15 K, pressure was set at 1 atm
and RH was fixed at 0.5%. Photolysis frequencies were calculated based on the cross sections,
quantum yields as described in Aumont et al. (2005) and the photonic flux of blacklight lamps.
Each simulation ran for 1 hour and the time evolution of species concentration were computed
through a two-step method that solves stiff ordinary differential equations (Verwer, 1994;
Verwer et al., 1996). To investigate effects of mass concentrations, we also simulated
experiments of n-alkane photooxidation under high $NO_x$ conditions with low mass loadings by
Presto et al. (2010). The number concentration of seed particles with particle diameter of 200
nm was ~5000 $cm^{-3}$, corresponding to the mass concentration of ~20 $\mu g\ m^{-3}$. Initial mixing
ratios of n-alkane and $NO_x$ were in the range of 3 – 99 ppb and 1 – 5 ppm, respectively, as
reported in Presto et al. (2010) and these conditions were applied in the model.
The box model accounts for mass transfer kinetics of organic species between gas and
particle phases. Partitioning follows Raoult's law at equilibrium and partitioning kinetics is
described by the gas-particle mass transfer coefficient with the Fuchs-Sutugin approach
(Seinfeld and Pandis, 2016). For the base case scenario, we fixed the mass accommodation
coefficient ($\alpha$) to be 1 based on molecular dynamics simulations (Julin et al., 2014), assuming
particles being low viscous liquids without kinetic limitations of bulk diffusion. To account for
potential kinetic limitations in viscous particles, we applied an effective mass accommodation
coefficient ($\alpha_{eff}$) that is a function of volatility and bulk diffusivity (Shiraiwa and Pöschl, 2021):
$$\alpha_{eff} = \alpha_s \frac{1}{1 + \frac{\alpha_s\ \omega\ C^0}{4\ D_b\ \rho_p} \frac{r_p}{5} \cdot 10^{-12} \frac{g\ cm^{-3}}{\mu g\ m^{-3}}} \qquad (1)$$
where $\alpha_s$ is the surface accommodation coefficient assumed to be 1, $\omega$ ($cm\ s^{-1}$) is the mean
thermal velocity of the organic compound in the gas phase, $r_P$ (cm) is the particle radius, $\rho_p$ (g
$cm^{-3}$) is the particle density, and $C^0$ ($\mu g\ m^{-3}$) is the pure compound saturation mass
concentration. $D_b$ ($cm^2\ s^{-1}$) is bulk diffusivity as simulated by conversion of viscosity as detailed



below. $\alpha_{eff}$ values are shown as a function of $D_b$ and vapor pressure $p^o$ in Fig S3a. We
accounted for a reversible gas-to-chamber wall partitioning of gases and assumed a fixed first-
order deposition rate constant of $5\times10^{-4}$ s$^{-1}$ based on experimental observations and previous
modeling studies (Krechmer et al., 2016; La et al., 2016; Lim and Ziemann, 2009b). A
desorption rate constant from wall to the gas phase was derived by using a parameter of
$C_w/M_w\gamma_w$ of 9 µmole m$^{-3}$ for n-alkanes and 120 µmole m$^{-3}$ for oxidation products based on
chamber observations (Matsunaga and Ziemann, 2010), as discussed in La et al. (2016).

The glass transition temperatures ($T_g$) of organic compounds were predicted by the

machine learning-based model tgBoost (Galeazzo and Shiraiwa, 2022) and the
parameterization based on elemental composition (Derieux et al., 2018; Li et al., 2020). The
implementation of the compositional parametrization into the GECKO-A box model was done
in Galeazzo et al. (2021) with a thorough description of all the equations, assumptions and steps
adopted for the implementation of this viscosity estimation method. In this study we
implemented tgBoost, a newly developed machine learning model for better prediction of $T_g$.
tgBoost is a powerful model that can discern compositional isomers by functionality and predict
the glass transition temperature of an organic compound $i$ ($T_{g,i}$) with an uncertainty of $\pm$ 18.3 K
using the canonical SMILES notation of a molecule (Galeazzo and Shiraiwa, 2022). We have
implemented a pipeline (i.e., gecko2vec) into GECKO-A to predict $T_g$ of compounds from the
chemical mechanism in a fast and computationally efficient manner. Gecko2vec executes three
main steps: first, it translates the IDs of the compounds of interest of the GECKO-A mechanism
into the respective canonical SMILES notations (translation step); second, it transforms the
canonical SMILES notations into the respective molecular embeddings (i.e., unique 300-
dimensional numerical representations of molecules; embedding step); and finally, the
pretrained tgBoost model and its weights are loaded and used to predict $T_g$ of each species
(prediction step). Within the box model, the $T_g$ of total SOA particles ($T_{g,org}$) resulting from the
combination of its organic component and water mixture is computed using the Gordon–Taylor
equation (Dette et al., 2014; Koop et al., 2011; Zobrist et al., 2008). $T_{g,org}$ can be converted to
viscosity based on the Vogel-Tammann-Fulcher approach (Derieux et al., 2018) and viscosity
is further converted into bulk diffusivity using the fractional Stokes-Einstein equation (Evoy et
al., 2019).

**Laboratory experiments.**

SOA particles were generated from OH oxidation of $C_8$-$C_{17}$ n-alkanes in a 5.9 m$^3$ Teflon

environmental chamber filled with clean air under high NO$_x$ conditions in the presence of non-



volatile dioctyl sebacate (DOS) seed particles, as described in detail elsewhere (Lim and
Ziemann, 2009b). Briefly, 1 ppm of n-alkane, 10 ppm of methyl nitrite, and 10 ppm of NO were
added to the chamber from a glass bulb, and ~200–400 µg m$^{-3}$ of seed particles were added
from an evaporation-condensation apparatus. Relatively high mass concentrations of seed
particles were used so that semi-volatile compounds would condense to particles, minimizing
vapor deposition to chamber walls (Zhang et al., 2014; Matsunaga and Ziemann, 2010).
Blacklights covering two of the chamber walls were then turned on for 60 min to form OH
radicals by methyl nitrite photolysis (Atkinson et al., 1981). The amount of n-alkane reacted
was measured by collecting Tenax samples before and after the experiment and analyzing by
gas chromatography with flame ionization detection (GC-FID). Aerosol volume concentrations
were measured using a scanning mobility particle sizer (Docherty et al., 2005) and converted
to an SOA mass formed using a density of 1.06 g cm$^{-3}$. SOA mass yields (mass of SOA
formed/mass of n-alkane reacted) were calculated from the measured SMPS mass (corrected
for particle wall loss using the ~20% h$^{-1}$ decay in mass after the lights were turned off) and the
GC-FID analyses. The SOA yields measured in these experiments were reported previously
(Lim and Ziemann, 2009b), but in light of a recent comparison of the accuracy of our SMPS
measurements with filter sampling the values reported here are higher by a factor of 1.24
(Bakker-Arkema and Ziemann, 2021). A temperature-programmed thermal desorption (TPTD)
method was also used to measure the composition and volatility of aerosol particles. Particles
were sampled directly from the chamber into a thermal desorption particle beam mass
spectrometer (Tobias et al., 2000), where they were formed into a beam inside an aerodynamic
lens, transported into a high vacuum chamber, and impacted on a copper rod vaporizer that was
coated with a non-stick polymer and cooled to –40°C. After sampling for 30 min, the vaporizer
was warmed by room air to –5°C and then heated at 2°C min$^{-1}$ to 200°C. Compounds desorbed
according to volatility and entered a quadrupole mass spectrometer, where they were ionized
by 70 eV electrons prior to mass analysis. In one recent n-hexadecane experiment, we also
measured the composition of nitrate, hydroxyl, carbonyl (ketone + aldehyde), carboxylic acid,
ester, and peroxide functional groups in SOA using derivatization-spectrophotometric methods,
with the amount of -CH$_2$- groups calculated by difference (Ranney et al., 2023). We note that
in that experiment the SOA yield measured by filter sampling was nearly identical to the one
we measured previously after applying the above correction.

**Results and discussion**
**SOA yields and viscosity.**





Figure 1(a) shows the measured yields of SOA generated from the oxidation of n-
alkanes ($C_nH_{2n+2}$; $n$ = 8 - 17) (Lim and Ziemann, 2009b). The model base case (black line) with
mass accommodation coefficient of 1 for all species represents no kinetic limitations in the
particle phase and the results are similar to previous simulations performed by La et al. (2016).
Vapor wall loss was considered based on experimental observations and previous modeling
studies (Krechmer et al., 2016; La et al., 2016; Lim and Ziemann, 2009b), which is important
to account for as no wall loss would lead to a significant overestimation of SOA yields, as
shown in the black dotted line and was discussed in detail in La et al. (2016). Both experimental
and simulated SOA yields increase with an increase of $n$, reflecting the decrease in volatility of
the precursor and its oxidation products (Shiraiwa et al., 2014). The observed SOA yield trend
is consistent with measurements by a thermal desorption particle beam mass spectrometer,
showing that n-alkane SOA are composed of less oxidized products with volatility lower for
precursors with higher $n$ (Lim and Ziemann, 2009b, a).
The overall good agreement suggests that multigenerational chemistry in the gas phase
and partitioning of semi- and low-volatile products, as explicitly treated by GECKO-A box
modeling, are the dominant pathway of n-alkane SOA formation under these conditions. It also
suggests that peroxy radicals ($RO_2^.$) mainly react with NOx, minimizing auto-oxidation and
gas-phase dimerization by $RO_2^. + RO_2^.$ reactions. Good model agreement also suggests minor
contributions from particle-phase oligomerization chemistry, while particle-phase unimolecular
reactions including cyclization of hydroxyketones and dehydration of cyclic hemiacetals
forming dihydrofurans are treated in the model as they are important for the further oxidation
due to the presence of a double bound in the dihydrofurans (Lim and Ziemann, 2009a; La et al.,
2016). Thus, the GECKO-A model seemingly treats all essential processes for simulations of
n-alkane SOA formation under high NOx conditions. Note that particle-phase chemistry was
shown to be substantial in n-alkane SOA formation under low NOx conditions through
peroxyhemiacetal and oligomer formation (Shiraiwa et al., 2013; Ziemann and Atkinson, 2012).
To explore the potential impacts of particle phase state on SOA formation and
partitioning, we implemented effective mass accommodation coefficient ($\alpha_{eff}$) that can
effectively consider kinetic limitations of bulk diffusion and also account for the effect of vapor
pressure on partitioning kinetics for species with various volatilities (Shiraiwa and Pöschl,
2021). Bulk diffusivity evolves upon SOA formation, which can be derived by viscosity and
glass transition temperature as predicted from the machine learning-based tgBoost model
(dashed green line) and the compositional parametrization (CP, dashed orange line). The
simulated SOA yields with tgBoost are very similar to the base case scenario with $\alpha$ = 1, while



the application of the CP leads to smaller SOA yields for $n$ = 15-17.  These results indicate that
$\alpha_{eff}$ is close to 1 with little kinetic limitations of bulk diffusion for most cases, except some
limitations are predicted by CP for large precursors. Deviations of tgBoost and CP stem from
the difference in phase state and viscosity predicted by the two methods.

Figure 2(a) shows the simulated viscosity and corresponding bulk diffusivity of n-

alkane SOA. Remarkably, the two models predict contrasting trends. The simulated glass
transition temperature ($T_{g,org}$) of SOA is presented in Fig. A1. The CP predicts a decrease in
$T_{g,org}$ for $C_{8-12}$ with the lowest $T_{g,org}$ of ~250 K followed by an increase of $T_{g,org}$ with $n$ to reach
~270 K with $C_{17}$. These values correspond to viscosity of $10^4$ - $10^6$ Pa s, indicating that n-alkane
SOA adopts viscous semisolid phase state. The increase of viscosity for larger precursors is
apparently reasonable, as their oxidation products would have higher molar mass which would
generally correspond to higher $T_{g,org}$ (Koop et al., 2011; Shiraiwa et al., 2017). Based on the
Stokes-Einstein relation, bulk diffusivity would be in the range of $3\times10^{-15}$ - $10^{-12}$ $cm^2$ $s^{-1}$. The
characteristic timescale of bulk diffusion in an average particle diameter of 300 nm can be as
low as ~2 hours (Shiraiwa et al., 2011), which is longer than experimental timescale of one
hour. These low diffusivities and long diffusion timescale can induce concentration gradients
in the particle bulk, reducing $\alpha_{eff}$ to cause significant kinetic limitations to retard SOA growth,
which is not consistent with the measured SOA yields.

Surprisingly, tgBoost predicts the opposite trend, predicting a monotonic decrease of

$T_{g,org}$ and viscosity with an increase of $n$, suggesting that SOA phase state shifts from an
amorphous semisolid state ($10^2 < \eta < 10^5$ Pa s) towards a liquid-like phase state ($\eta < 10^2$ Pa s).
These results are counter-intuitive as $T_g$ values of n-alkanes increases with an increase of $n$,
which can be reproduced with great precision by tgBoost (Galeazzo and Shiraiwa, 2022). The
determinants explaining this unexpected trend are chemical composition and molecular
structure of the oxidation products as discussed below. The characteristic timescale of bulk
diffusion is less than one second in a low viscous state and high bulk diffusivity (Shiraiwa et
al., 2011) and SOA particles are expected be homogeneously well-mixed. Hence, $\alpha_{eff}$ remains
very close to 1 with little kinetic limitation of bulk diffusion.

Unfortunately, no direct viscosity measurements of n-alkane SOA generated under high

$NO_x$ conditions are available to date, while there are two studies for n-alkane SOA generated
under NOx-free conditions. Saukko et al. (2012) (Saukko et al., 2012) observed that n-
heptadecane ($C_{17}H_{36}$) SOA with low O:C ratio did not bounce from an impactor plate. It
indicates that these particles adopted a liquid-like state, as indicated by the violet shading in





Fig. 2(a), which is consistent with the tgBoost prediction. Shiraiwa et al. (2013) estimated bulk
diffusivity of n-dodecane ($C_{12}H_{26}$) SOA generated without $NO_x$ to be $10^{-12}$ cm$^2$ s$^{-1}$ using a
kinetic multilayer model to simulate evolution of particle size distribution. While these two data
points cannot be directly compared with the viscosity predictions of high NOx n-alkane SOA,
they serve as reference data points for now and direct viscosity or bulk diffusivity
measurements of high $NO_x$ n-alkane SOA are warranted in future studies.

Figure 2(b) shows the thermal desorption profiles of DOS that was present as seed

particles within the SOA formed from oxidation of the n-alkanes. Since DOS desorption
involved diffusion through the SOA prior to escape into vacuum, these profiles provided a
means for probing the SOA viscosity. The peaks in the DOS profiles for the $C_{8-13}$ and $C_{14-17}$ n-
alkanes are closely grouped, with vaporizer temperature at ~80 °C and ~65 °C, respectively,
with the peak for pure DOS occurring in between at ~72ºC. The observed decrease in desorption
temperatures from low to high carbon numbers suggests an increase in effective volatility of
DOS in SOA generated from larger n-alkanes. In addition, Lim and Ziemann (2009) have
observed that $C_{10}$ n-alkane SOA generated under high NOx conditions evaporate at higher
temperatures compared to $C_{12}$ and $C_{15}$ n-alkane SOA based on total ion thermal desorption
measurements (Lim and Ziemann, 2009b). Volatility and $T_g$ were shown to exhibit clear
anticorrelation (Li et al., 2020); hence, these results strongly indicate that $C_{8-13}$ SOA have higher
$T_g$ and viscosity compared to $C_{13-17}$ SOA.  It is remarkable to note that the $C_{13}$ profile is bimodal
with peaks at ~80 °C and ~65 °C (Fig. 2b), which is in line with tgBoost prediction that the
viscosity of $C_{13}$ alkane SOA is at the edge of amorphous semi-solid and liquid phase states (Fig.
2a). These results indicate that n-alkane SOA generated by larger precursors adopt low viscous
liquid-like states, while n-alkane SOA generated by smaller precursors adopt viscous semisolid
states, in agreement with tgBoost predictions. The major strength of tgBoost is that it considers
molecular structure and functionality for $T_g$ predictions, while the compositional
parameterization does not account for this effect, leading to intuitive but erroneous predictions.

**Chemical composition of SOA.**

Figure 1 also shows the simulated (b) N:C and (c) O:C ratios of SOA with $\alpha = 1$ (black

line) and $\alpha = \alpha_{eff}$ with $T_g$ determined with tgBoost (green line) or the compositional
parameterization (orange line). The N:C ratio is very similar among all simulations being ~0.2
for $C_8$ and decreasing progressively to ~0.03 with each addition of a carbon atom in the
precursor. O:C ratios were calculated in two different ways by treating a nitrate (-$ONO_2$) group



to contain either three (solid lines) or one (dashed lines) oxygen atoms. One oxygen atom is
also considered because O:C ratios reported from aerosol mass spectrometer measurements
generally treat a nitrate group the same as a hydroxyl group, since they have the same effect on
oxidation state (Farmer et al., 2010). Similar to the N:C ratio, there is a constant decrease in
O:C of SOA with increasing $n$, which is consistent with previous measurements for n-
pentadecane ($C_{15}H_{32}$) SOA (Aimanant and Ziemann, 2013a) and n-hexadecane ($C_{16}H_{34}$) SOA
in this study, even though the simulated values are ~45% and 15% lower than the measured
N:C and O:C ratios, respectively.
We measured functional group distributions in n-hexadecane SOA using derivatization-
spectrophotometric methods described in Aimanant and Ziemann (2013b), as shown in Fig.
1(d) and summarized in Table A1. Experimental measurements report high presence of -$CH_2$-
(13.81) and -$ONO_2$ (0.91), followed by ROH (0.41), RC(=O) (0.38), and RC(=O)OR (0.28),
with the average measured number of groups per $C_{16}$ molecule in parenthesis. Figure 1(d)
includes simulation results by GECKO-A with CP and tgBoost, showing overall satisfactory
agreement. The simulated results with tgBoost show excellent agreement for hydroxyl and
methylene groups, while the simulated nitrates and carbonyls (ketones + aldehydes) are lower
than the measurements. The simulation by CP has also a similar trend, but with significantly
lower presence of nitrates, carbonyls, and esters.
Figure 3(a) shows the top 15 oxidation products in the particle phase formed by the
oxidation of n-hexadecane simulated by GECKO-A box model with tgBoost. Note that
positional isomers are lumped into one species and five species in the first row constitute
majority (~86%) of SOA mass. The simulated SOA is composed mostly by 1st generation
products including alkyl nitrates, hydroxynitrates, and hydroxyketones. There is also a
significant presence of 2nd and 3rd generation products such as esters and dinitrates. We also
found multi-functionalized decomposition products including smaller chain hydroxy nitrates
and alkyl lactones as well as particle-phase products from cyclization of hydroxyketones and
dehydration of cyclic hemiacetals to form dihydrofurans. There are notable differences in
molecular composition for SOA simulated by CP (Fig. A2): the major compounds are 1st
generation single and multi-functionalized products, followed by some 2nd and 3rd generation
products, without decomposition products in the top species.
The simulated $T_g$ by both methods for each compound are listed in Fig. 3. Overall
tgBoost predicts $T_g$ to be 157 – 221 K which are much lower compared to CP, especially with
significant differences for organic nitrates and multi-functionalized species. As tgBoost
considers the molecular structure, functional group and atomic interconnectivity of a molecule,



it should make better predictions for multi-functionalized compounds based on the presence of
different functional groups. CP is based on elemental composition and it predicts high $T_g$ for
compounds with high molar mass, predicting same $T_g$ for isomers. In addition, the CP for
CHON compounds was developed based on $T_g$ values mainly estimated from their melting
points, as there are limited number of CHON compounds with measured $T_g$ available. $T_g$ of
organic nitrates are especially scarce and future $T_g$ measurements for organic nitrates are desired
to improve $T_g$ parameterizations. For these reasons, CP overestimates $T_g$ for oxidation products
of n-alkane with long chain on average by ~66 K compared to tgBoost, overpredicting SOA
viscosity as shown in Fig. 2(a).

Figure 3 also lists $\alpha_{eff}$ values, showing that they are very close to 1 for tgBoost, with

SOA to be low viscous liquid with little kinetic limitations in mass accommodation. Additional
oxidation products with lower concentrations are listed in Fig. A3 and their $\alpha_{eff}$ remain also
close to 1. In contrast, as CP predicts SOA phase state to be viscous amorphous semisolid, $\alpha_{eff}$
values for semi-volatile compounds become significantly smaller to kinetically limit mass
accommodation. This decrease of $\alpha_{eff}$ is larger for compounds with higher volatility, as such
compounds have higher re-evaporation rate on viscous particles with lower rate of bulk
diffusion (Shiraiwa and Pöschl, 2021) (Fig. A3). $\alpha_{eff}$ for lower volatility compounds remain
high, as they exhibit much lower desorption rates and are less likely to re-evaporate, even if
their diffusion into the bulk is slow. Consequently, SOA simulated with CP mainly consists of
later generation products with higher functionalization and molar masses.

Figure 3(b) shows top 15 oxidation products of n-decane ($C_{10}H_{26}$) as predicted by

GECKO-A with tgBoost. SOA is mostly composed of 2nd and 3rd generation products with
multiple functional groups including nitrates, ketones, and alcohols. These highly oxidized
products have $T_g$ in the range of 225 – 304 K, with similar predictions by CP and tgBoost. This
is consistent with previous studies that demonstrated successful applications of CP to predict
the measured viscosity of SOA derived from biogenic and other relatively small precursors
(Derieux et al., 2018; Smith et al., 2021; Baboomian et al., 2022). These results are consistent
with total ion thermal desorption profiles of n-alkane SOA formed in the presence of NOx (Lim
and Ziemann, 2009b): $C_{10}$ SOA was observed to have a broad single peak around ~75 °C,
indicating the presence of low volatility multigenerational products; in contrast, $C_{12}$ and $C_{15}$
SOA exhibited two peaks with one larger peak at lower temperature, corresponding to 1st
generation products and another smaller peak for multigenerational products. The phase state





of n-decane SOA is predicted to be semisolid, but kinetic limitations are not strong as $\alpha_{eff}$ values
for most compounds are only slightly reduced from 1.

**Effects of mass loadings on viscosity.**

The use of higher mass loadings in chamber experiments than ambient conditions

assured that the condensation of semi-volatile vapors to suspended particles is a dominant
process over vapor wall deposition (Zhang et al., 2014; Matsunaga and Ziemann, 2010).
Chamber experiments of n-alkane photooxidation at high $NO_x$ were also conducted with lower
mass loading by Presto et al. (2010). As shown in Fig. 4(a), SOA yields are increased with an
increase of SOA mass concentrations , which is in consistent with SOA absorptive partitioning
theory (Pankow, 1994). The oxidation of larger precursors leads to higher SOA yields, in
agreement with Lim and Ziemann (2009b) as presented in Fig. 1a. As shown with solid lines,
the GECKO-A box model simulated experimental observations very well.

Figure 4(b) depicts the simulated SOA viscosity. We observed the same trend as Fig.

2(a) with lowering of viscosity upon an increase of carbon number $n$. SOA phase state is
predicted to be semisolid for low carbon $n$, while it is expected to be liquid for high $n$. The
predicted viscosity is about one order of magnitude higher compared to Fig. 2(a). Lower mass
loadings suppress partitioning of higher volatility compounds, resulting in higher viscosity as
condensation would be dominated by lower volatility compounds with higher $T_g$ (Jain et al.,
2018; Champion et al., 2019; Grayson et al., 2016; Derieux et al., 2018).

**Atmospheric Implications.**

The phase state and viscosity of SOA formed by IVOCs have been largely unknown

and unexplored. We demonstrated in this study that SOA derived from small and middle size
n-alkane ($C_{12}$ and smaller) mostly consists of multigenerational oxidation products to adopt an
amorphous semisolid state, while larger n-alkane SOA are mainly composed of first generation
lightly oxidized products to adopt a low viscous liquid state. This result is surprising and
counterintuitive, as it has been established that higher molar mass would lead to higher glass
transition temperature, and hence, higher viscosity (Koop et al., 2011; Shiraiwa et al., 2017). In
fact, the viscosity of biogenic SOA follows this trend: the viscosity of isoprene ($C_5H_8$) SOA is
reported to be lower than monoterpene ($C_{10}H_{16}$, such as $\alpha$-pinene and limonene) SOA
(Renbaum-Wolff et al., 2013; Zhang et al., 2019), while oxidation products of sesquiterpene
($C_{15}H_{24}$) increase viscosity of SOA (Smith et al., 2021), which is captured by empirical
parameterizations based on elemental composition (Derieux et al., 2018; Li et al., 2020). In



contrast, n-alkane SOA exhibits an opposite trend, as indicated by thermal desorption
measurements that show that DOS in SOA formed by oxidation of large n-alkanes has higher
volatility. Hence, the SOA has lower viscosity, due to the enhanced presence of less
functionalized first-generation products (Li et al., 2020; Zhang et al., 2019). This trend is
successfully predicted by GECKO-A combined with machine learning-based model tgBoost,
which emphasizes the importance of consideration of functionality and molecular structure in
accurate predictions of $T_g$. The relationship between viscosity and composition is also reflected
in the atomic O:C and N:C ratios of n-alkane SOA, which decrease monotonically upon an
increase of carbon number of the n-alkane, since higher oxidation state and functionalization
can increase $T_g$ (Derieux et al., 2018; Koop et al., 2011; Shiraiwa et al., 2017; Saukko et al.,
2012).

IVOCs gain growing attention for better characterization of urban air quality, as they
represent an important source of SOA as shown by chamber experiments (Aimanant and
Ziemann, 2013a; Lim and Ziemann, 2009b) and as observed in field observations (Gentner et
al., 2012; Li et al., 2022; Robinson et al., 2007; Mcdonald et al., 2018). While a few large-scale
aerosol models treat IVOC SOA to achieve better agreement with ambient measurements (De
Gouw et al., 2011; Li et al., 2022; Zhao et al., 2016), IVOC SOA is still highly uncertain in
terms of chemical composition and particle phase state and model parameters and treatments
for SOA formation and partitioning are poorly constrained. Our study provides critical insights
for these aspects, showing that n-alkane SOA formation under high NOx conditions (as usually
the case for ambient urban air) is dominated by gas-phase chemistry followed by partitioning.
As the generated SOA particles adopt a low viscous state, there is little kinetic limitations of
mass accommodation and bulk diffusion, which supports the application of equilibrium SOA
partitioning in the boundary layer. While the experiments and modeling were conducted for dry
conditions in this study, the phase state and viscosity of ambient n-alkane SOA would be
expected to be even lower under humid conditions due to hygroscopic growth and water acting
as plasticizer. Note that further experiments and model simulations are required for different
conditions for middle and upper free troposphere, as viscosity is expected to become higher
under low temperatures.
It is highly remarkable that the combination of tgBoost and GECKO-A box model
accurately simulates SOA yields, functional group distributions and phase state. This new
model represents a unique and comprehensive tool for simulating formation, partitioning and
chemical evolution of SOA, opening up a new avenue for analyzing complex interplay of gas-
phase chemistry and particle-phase processes and composition in SOA for detailed analysis and



interpretation of laboratory experiments and field observations. In addition, we propose to
pursue the application of this model as a basis for the development of a detailed master
mechanism of multiphase aerosol chemistry as well as for the derivation of simplified but
realistic parameterizations for air quality and climate models. In regional and global air quality
models, it is challenging and computationally very expensive to treat complex SOA multiphase
processes. Thus, such processes should be treated in efficient but effective way and the new
model shall serve as benchmark for the development of simplified SOA descriptions.


**Acknowledgements.** This work was also funded by U.S. Department of Energy (DE-
SC0018349), U.S. National Science Foundation (AGS-1654104) and the Campus France
(Make Our Planet Great Again short stay program grant, mopga-short-0000000116). In
addition, PZ acknowledges support from the National Science Foundation under grant AGS-

467    1750447.


**Authors contributions.** TG and MS designed the study. TG conducted model simulations and
data analysis. RV, MC, and BA developed the GECKO-A model. YL and PZ conducted
experimental measurements. All authors discussed the results. TG and MS wrote the manuscript
with contributions from all coauthors.
**Competing interests.** At least one of the (co-)authors is a member of the editorial board of
Atmospheric Chemistry and Physics.
**Code/Data availability.** The simulation data may be obtained from the corresponding author
upon request. The model tgBoost is available in Github (https://github.com/U0M0Z/tgpipe) and
in the homepage (https://azothai.ps.uci.edu/).

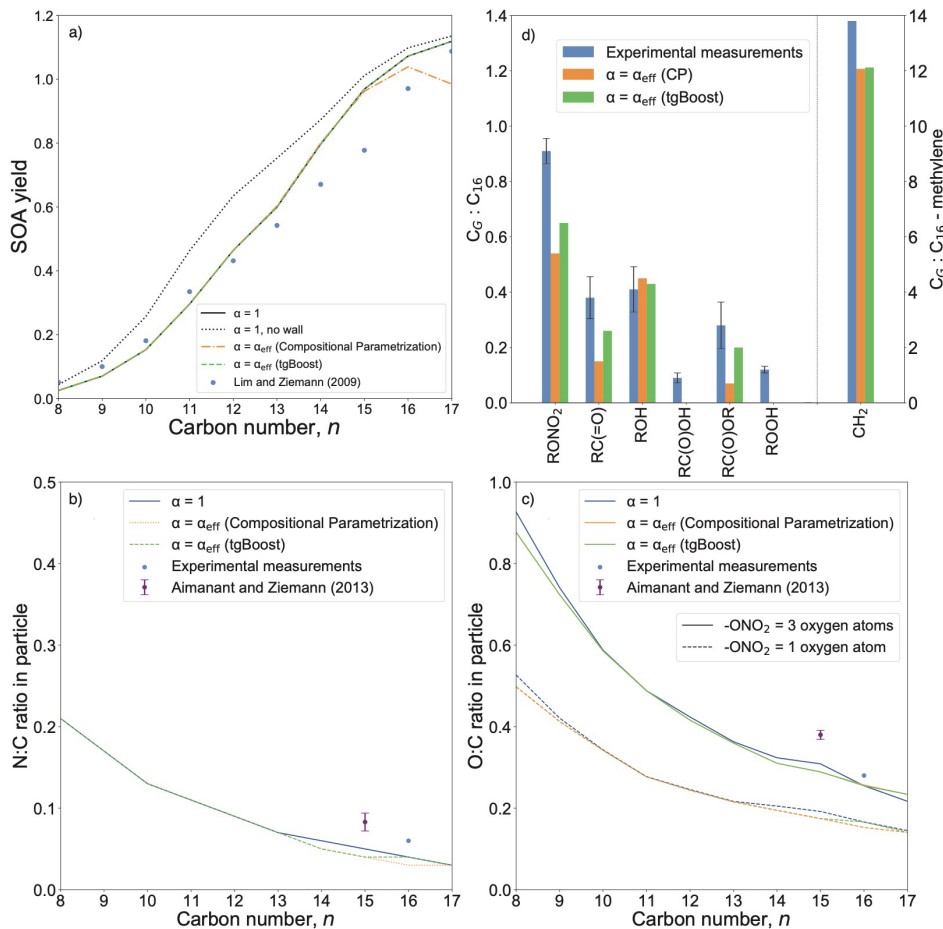

**Figure 1:** (a) Yields of SOA generated from OH oxidation of linear n-alkanes as measured by Lim and Ziemann (2009) (markers) (Lim and Ziemann, 2009b) and modeled by the GECKO-A box model (lines). The black line represents the base case with mass accommodation coefficient ($\alpha$) of 1. The dashed lines represent simulations with effective mass accommodation coefficient ($\alpha_{eff}$) as a function of bulk diffusivity from tgBoost (green) and the compositional parameterization (orange). (b) N:C and (c) O:C ratios in SOA formed by n-alkane oxidation simulated by the GECKO-A box model. The black line represents the base case with $\alpha$ of 1. The dashed lines represent simulations with $\alpha_{eff}$ with tgBoost (green) and the compositional parameterization (orange). (d) Simulated functional group distributions of n-hexadecane ($C_{16}H_{34}$) oxidation products in the particle phase. The blue bars represent experimental measurements. The green and orange bars represent GECKO-A box model simulations with $\alpha_{eff}$ with tgBoost and the compositional parameterization, respectively.





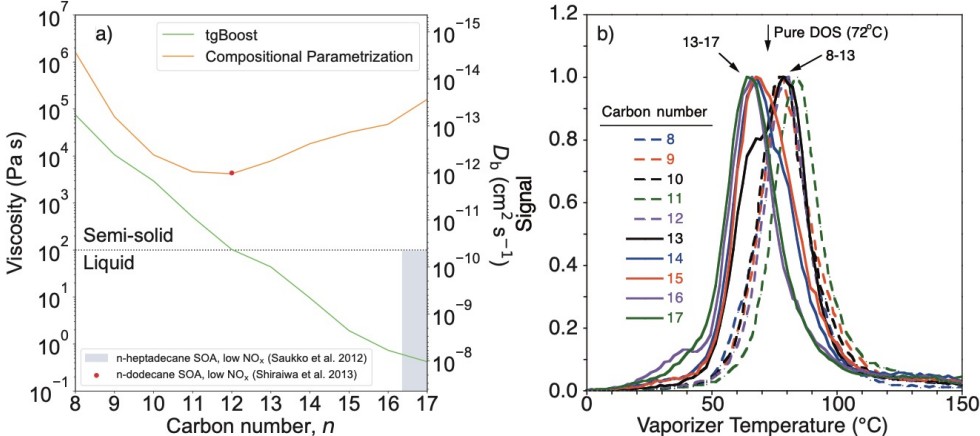


**Figure 2:** Phase state of n-alkane SOA. (a) Predicted viscosity of SOA generated from n-
alkanes as computed by the GECKO-A box model with the $T_g$ compositional parametrization
(orange line) and tgBoost (green line) at the last step of the simulations (t = 3600 s). (b) Thermal
desorption temperatures of n-alkane SOA ($C_{8-17}$) formed on dioctyl sebacate (DOS) seed
particles under high NOx conditions.






**Figure 3:** Molecular composition of oxidation products of n-alkanes under high NOx conditions in the particle phase. Top 15 SOA contributors with highest concentrations in (a) n-Hexadecane ($C_{16}H_{34}$) SOA and (b) n-Decane ($C_{10}H_{32}$) simulated by GECKO-A with effective mass accommodation coefficient ($\alpha_{eff}$) with tgBoost. The species are reported in descending concentrations from left to right and from top to bottom. Positional isomers are lumped into one species. Listed values are $T_g$ as calculated by tgBoost and CP and $\alpha_{eff}$ values at the end of simulation (3600 s) in brackets. Types of compounds are also noted (1st, 2nd, and 3rd generation products, decomposition products).





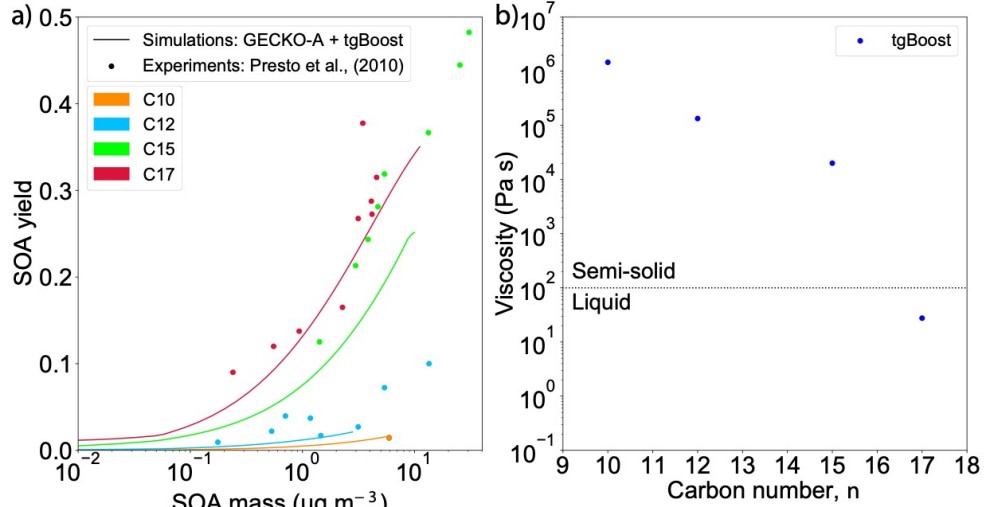

**Figure 4:** Effects of mass loadings on SOA yields and viscosity. (a) SOA yields from photo-oxidation of n-decane (C10), n-dodecane (C12), n-pentadecane (C15), and n-heptadecane (C17) at high NOx as a function of SOA mass concentration, as measured in Presto et al. (2010) (markers) and as modeled by the GECKO-A box model combined with tgBoost (lines). (b) SOA viscosity as modeled by the GECKO-A box model combined with tgBoost.



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



**Appendix.**

**Table A1:** Experimental and simulated functional group distributions, O:C and N:C ratios of SOA generated from C16 oxidation by OH in presence of high $NO_x$.

| FG/C16 molecule | Experimental | Simulated (tgBoost) | Simulated (CP) |
|---|---|---|---|
| Nitrate | 0.91 | 0.65 | 0.54 |
| Carbonyl | 0.38 | 0.26 | 0.15 |
| Hydroxyl | 0.41 | 0.43 | 0.45 |
| Carboxyl | 0.09 | 0.0 | 0.0 |
| Ester | 0.28 | 0.2 | 0.07 |
| Peroxide | 0.12 | 0.01 | 0.0 |
| Methylene | 13.81 | 12.12 | 12.07 |
| O:C | 0.28 | 0.25 | 0.25 |
| N:C | 0.06 | 0.04 | 0.03 |
| H:C | 1.85 | / | / |
| MW | 294 | / | / |
| Density (g cm$^{-3}$) | 1.10 | 1.06 | 1.06 |

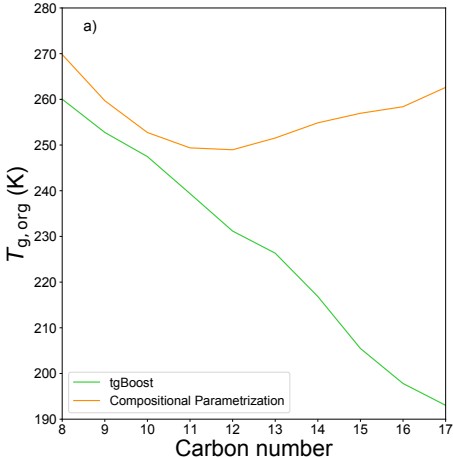

**Figure A1:** Predicted $T_{g,org}$ of SOA generated from n-alkanes as computed by the GECKO-A box model with the $T_g$ compositional parametrization (orange line) and tgBoost (green line) at the last step of the simulations (t = 3600 s).



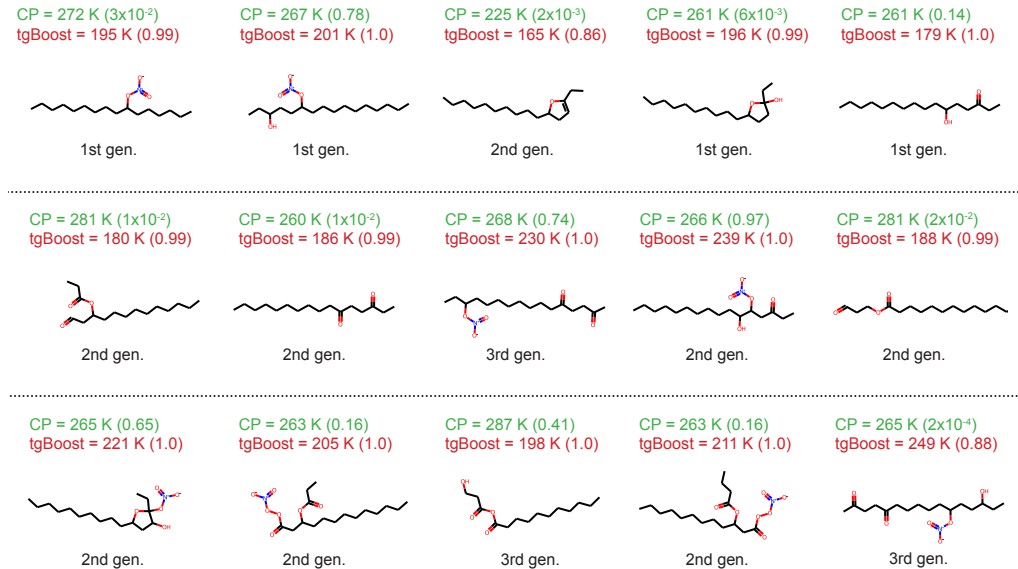

**Figure A2.** Top 15 species with highest concentrations in oxidation products of n-hexadecane ($C_{16}H_{34}$) under high NOx conditions simulated by GECKO-A with effective mass accommodation coefficient ($\alpha_{\text{eff}}$) with the compositional parameterization. The species are reported in descending concentrations from left to right and from top to bottom. Listed values are $T_g$ as calculated by tgBoost and CP and $\alpha_{\text{eff}}$ values at the end of simulation (3600 s) in brackets. Types of compounds are also noted (1st, 2nd, and 3rd generation products, decomposition products).



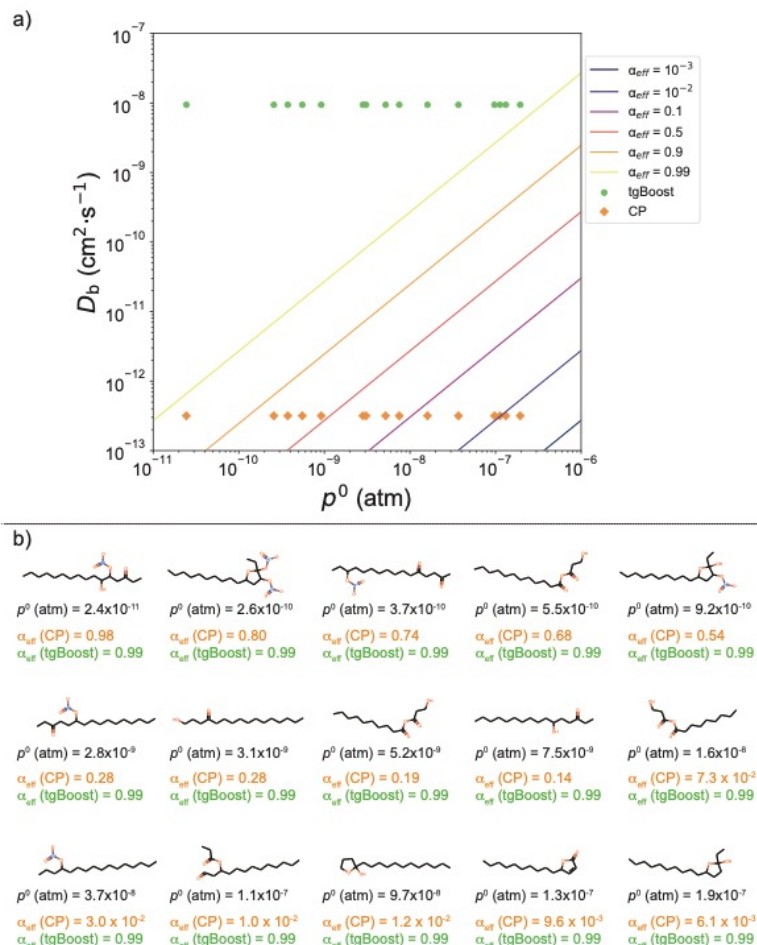

806

**Figure A3.** a) $\alpha_{eff}$ isolines as a function of bulk diffusivity $D_b$ and saturation vapor pressure $p^o$ of semi-volatile species. b) Selection of various representative SOA contributors produced during the oxidation of n-hexadecane. The species are ordered by decreasing vapor pressure. The reported $\alpha_{eff}$ values for each SOA contributor are calculated for $D_b$ estimated with tgBoost ($D_b = 1\times10^{-8}$ cm$^2$ s$^{-1}$) and CP ($D_b = 3\times10^{-13}$ cm$^2$ s$^{-1}$). The values of $\alpha_{eff}$ for the selected species are reported as points in the top panel. It shows that for the liquid-like state estimated with the tgBoost configuration, $\alpha_{eff}$ tend towards 1 for all species. This behavior is not observed in the amorphous semi-solid state estimated using the CP model configuration for species with $p^o$ above $10^{-9}$ atm. For the simulated conditions, species with $p^o$ between $10^{-8}$ and $10^{-6}$ atm are of enough low volatility to partition between the particle and gas phases at equilibrium. For species in that volatility range, no mass transfer limitation is observed with the tgBoost configuration, unlike the CP configuration. Using the CP configuration, the most volatile SOA contributors are subjected to substantial mass transfer limitation and are therefore mainly eliminated by gas-phase oxidation or wall deposition.

821