# Peer review of "Secondary Organic Aerosols Derived from Intermediate Volatility"

_EGUsphere, 2024_

## Author Comment (AC1)

**Response to Referee 1** (comments in black, response in blue)

In the manuscript " Secondary Organic Aerosols Derived from Intermediate Volatility n-Alkanes Adopt Low Viscous Phase State, " the authors reported that n-alkane SOA with higher carbon number mainly consists of less functionalized first-generation products with lower viscosity, while the lower carbon number SOA contains more functionalized multigeneration products with higher viscosity based on the GECKO-A box model simulation and chamber experiments. In general, I am very supportive of the hypothesis that the increase of the alkyl group may reduce the viscosity of SOA from n-alkanes, and the topic of this work is interesting to readers of this subject.

We thank Referee 1 for the review and positive evaluation of our manuscript.

Only the functional group information was provided in this study, which can indeed help understand SOA's total properties. However, since there is no specific molecular information in the chamber data, it is very difficult to evaluate the mechanisms generated by the GECKO-A. Using the GECKO-A box model isn't helpful unless you know which products are really forming in the gas phase and subsequent particle-phase chemistry in the aerosol. Additionally, the mechanism's performance should be carefully characterized by chamber experiments before it is used to evaluate the molecular properties of SOA. Thus, for the publication of this manuscript, the following points should be addressed.

In this study, the modeled results were compared against all measured data available for the selected experiments on n-alkane oxidation at high-NOx, including SOA yield, O:C ratio, and functional group distributions. A very recent study by Ranney et al. (2023) measured specific molecular information of n-hexadecane oxidation products, detecting alkyl nitrates, hydroxyl nitrates, hydroxyl carbonyls, cyclic hemiacetals, and cyclic hemiacetal nitrates as major products. These compounds are among top 15 SOA contributors as shown in Fig. 3a. This agreement is strong indications that GECKO-A simulates molecular products well. We add this point in the revised manuscript. We would like to point out that most previous SOA modeling studies simulate and compare just SOA yield and O:C ratio, and the model comparison with functional group distributions and N:C ratio in this study is highly unique, as they are often not measured nor modeled. The GECKO-A model is very powerful, as it is one of the only SOA models which resolves gas-phase chemistry explicitly to enable comparison with the measured functional group distributions. We also note that measurements of particle-phase concentrations of specific molecular products are very challenging. Mass spectrometry measurements provide elemental composition without resolving functional groups nor isomers, and one requires standard compounds to robustly quantify concentrations, which are often unavailable for SOA oxidation products.

Line 19, in the abstract section, the authors stated that SOA derived from n-alkanes is the dominant component of anthropogenic particulate matter; however, it is generally believed that aromatics and alkenes are the main precursors of anthropogenic aerosols.

Following your comment, we change the word dominant to major. As cited in the manuscript, there are plenty of studies showing that alkane SOA are major components of anthropogenic SOA: anthropogenic precursors are mostly composed of alkanes (40%), followed by aromatics (20%)

and alkenes (10%) with the rest being oxygenated and unidentified compounds (Ziemann and Atkinson, 2012; Shrivastava et al., 2022). We have added these references in the revised manuscript.

In the methods section, the authors showed that GECKO-A generated the chemical mechanisms of n-alkanes; however, the details about the mechanisms were not provided.

The detailed protocol for mechanism generation for n-alkane oxidation mechanism is available in previous studies (Aumont et al., 2013; Aumont et al., 2005; Aumont et al., 2012; La et al., 2016). We have added more information in the revised manuscript as below:

"The GECKO-A generator used for oxidation of linear n-alkanes treats chemistry of peroxy ($RO_2$) and alkoxy (RO) radicals. Under high NOx conditions, $RO_2$ radicals mainly react with NO and $NO_2$, to form closed-shell compounds or RO radicals, which undergo reaction with $O_2$, unimolecular decomposition (i.e. C-C bond breaking) or isomerization, which generate stable compounds and/or to new $RO_2$ radicals. The detailed protocol for such mechanism generation is available in previous studies (Aumont et al., 2013; Aumont et al., 2005; Aumont et al., 2012; La et al., 2016)."

Line 123: How many species were involved during the SOA formation process? Detailed information about the box model should be provided.

We run the mechanism up to four generations. The breakdown of species per chemical mechanism is: C8 = 12686, C9 = 23041, C10 = 36109, C11 = 52460, C12 = 73565, C13 = 92909, C14 = 118668, C15 = 136569, C16 = 141519, C17 =159640. We think that such exact number of species is not essential, so we included the following sentence in the revised manuscript.
"Species with vapor pressure below $10^{-13}$ atm are assumed to be of low enough volatility to completely partition to the condensed phase and their gas phase chemistry is then not generated in the mechanism to reduce the mechanism (La et al., 2016). The number of species treated in the model was ~$10^4$ species for dodecane ($C_8H_{18}$) that increases to ~$10^5$ species for heptadecane ($C_{17}H_{36}$)."

In line 233, the authors only compared the yields of SOA between chamber data and model simulations; without further comparisons of chemical compositions, it is hard to conclude that particle-phase oligomerization contributes minor. Additional information or references are needed to support this statement.

First, we point out that, in addition to SOA yields, we also compared O:C and N:C ratios as well as functional group distributions (Fig. 1). In addition, the modeled viscosity was validated against thermal desorption measurements (Fig. 2b). Recently, Ranney et al. (2023) suggested that cyclic hemiacetals form acetal dimers in the particle phase for SOA formed from the reaction of n-hexadecane SOA and OH/NOx. While they could not directly detect dimers, their derivatization measurements indicate the presence of acetal dimers. In the revised manuscript, we have toned down our statement to mention that oligomerization chemistry is not a dominant process and then mention these latest results by Ranney et al. (2023). We also note that the impact of such particlephase chemistry may warrant further investigations by future model development and experimental studies.

Line 261, why does the $T_{g,org}$ decrease for $C_{8-12}$ in Figure 2a? Some explanation should be provided.

This decrease for $C_{8-12}$ is likely due to steep decrease of O:C ratio upon an increase of $n$ as shown in Fig. 1d, as lower O:C ratio can lead to a decrease of $T_g$ (Shiraiwa et al., 2017; DeRieux et al., 2018). We clarify this point in the revised manuscript.

Figure 2b, some semi-volatile compounds may escape from particles into the gas phase in the aerodynamic lens due to the high vacuum; hence, $T_g$ may be overestimated, and the effect of vacuum on SOA compositions should be discussed.

There seems to be misunderstanding. We did not use information in Fig. 2b to estimate $T_g$, but Fig. 2 was used to validate the predicted phase state based on GECKO-A modeling. Please note that Fig. 2b represents thermal desorption temperatures of dioctyl sebacate (DOS, vapor pressure is ~$2.4 \times 10^{-10}$ atm) that was present as seed particles in n-alkane SOA and it does not show thermal desorption temperatures of SOA itself. Compounds with vapor pressure < $10^{-5}$ Torr ($1.3 \times 10^{-8}$ atm) is estimated to undergo negligible evaporation as the residence time in the aerodynamic lens is only ~0.2 s (Tobias et al., 2000). Thus, DOS and most of SOA products should not be affected, while some volatile oxidation products may undergo evaporation. We revised and clarified this point in the revised method and caption of Fig. 2.

In Line 397, It shows that the model performs well on SOA yields by comparing the chamber and the GECKO-A box model; still, the time profiles of SOA mass concentrations should be provided to evaluate the performance of the box model.

Actually, Fig. 4a represents time profiles, as SOA mass concentrations (x-axis) evolves as a function of time and SOA yields (y-axis) are time-dependent: the data points in this figure were measured at different reaction time in experiments and the modeled lines represent temporal evolution of SOA yield (please see also description in Presto et al., 2010). Good agreement between measurements and simulations confirms good performance by the model. We clarify this point in the revised manuscript.

---

## Author Comment (AC2)

**Response to Referee 2**

**GENERAL COMMENTS:**
In this study, the authors predict the viscosity of n-alkane SOA under high NOx conditions using a box model (GECKO-A) combined two different Tg-predictors: the compositional parameterization (CP) and a machine-learning based approach (tgBoost). This paper is within the scope of ACP, and the results are relevant and useful for the atmospheric chemistry community in terms of better understanding the various trends of SOA viscosity. This study provides a clear benchmark for future work to test the viscosity predictions of tgBoost and CP to further validate the author's conclusions. However, some parts of the paper need further explanation and I also suggest rewording some parts of the paper for clarity, as discussed below. Overall, this paper is acceptable for publication in ACP after revision of the below points.

We thank Referee 2 for the review and positive evaluation of our manuscript.

**SPECIFIC COMMENTS:**
The authors point to previous studies that describe in detail the limitations and assumptions of using tgBoost and CP. This is normal practice in general, but it would strengthen the paper to add a brief summary of the main limitations and assumptions of each model here. The authors do mention the types of reactions that tgBoost does not explicitly consider, but nothing about other assumptions made for tgBoost or for combining GECKO-A with CP. A short discussion will add some needed context for the readers.

The difference between CP and tgBoost is consideration of molecular structure and functional group, as described in the manuscript. Box model simulations with CP and tgBoost were conducted with same conditions. For both model simulations with CP and tgBoost, the particle number concentration is assumed to remain constant (coagulation is not treated), while the particle radius evolves following the partitioning of organics. Potential concentration gradients in the particle phase are not resolved and SOA particles are assumed to be homogeneously well-mixed implicitly. Beyond these points, we do not have other assumptions for combining GECKO-A with tgBoost or CP. We have clarified these assumptions in the revised manuscript.

More details need to be provided for the calculation of viscosity and bulk diffusivity from Tg,org. What parameters were used in the VFT equation to calculate viscosity (e.g., what was Df?). What parameters were used in the fractional Stokes-Einstein equation to calculate bulk diffusivity (e.g., what hydrodynamic radii was used for the n-alkane SOA)? Please at least provide the parameters for this calculation. This could either go in the main text or supplement.

The fragility parameter ($D_f$) was assumed to be 10 based on DeRieux et al. (2018). An effective molecular radius was assumed to be 0.5 nm and a fractional SE parameter of 0.93 was used based on Evoy et al. (2019). We clarify these values in the revised manuscript.

In many cases, the authors describe their results as "counter-intuitive", "surprising", and "remarkable". I don't necessarily think these kinds of superlatives are appropriate in this case. From what I can tell, the authors describe three things as "remarkable": 1) that tgBoost and CP don't agree on the viscosity trend (e.g. Line 259, 272), 2) that the viscosity of n-alkanes decrease

with increasing n (Line 275), and 3) that tgBoost predicts the viscosity very well (Line 449). For points (1) and (3), the authors themselves state on Lines 352-355 that "as tgBoost considers molecular structure…. It should make better predictions for multi-functionalized compounds," which I agree with – tgBoost seems like a more sophisticated model, so I think it's reasonable that it would both perform better than CP and perform quite well when comparing to measurements. For point (2), I believe it's relatively well-hypothesized that SOA viscosity is not only dependent on molecular weight, but also functionality and structure. I suggest either revising or removing this type of language throughout.

As discussed in the manuscript and summarized by this comment, the results on lower viscosity from larger alkane precursors are counter-intuitive given that the molar mass has primary importance on determining $T_g$. In this sense, we were surprised by our results, but better performance by tgBoost should not be surprising. Hence, we remove the word "surprising" throughout the revised manuscript and removed "counter-intuitively" from abstract to tone down.

The authors mention studies where CP viscosity predictions have agreed with viscosity measurements (Line 378-380), but if I recall the literature correctly, there have been previous studies where CP (i.e., the DeRieux et al., 2018 paramterization) has not agreed with experimental viscosities. Mentioning some of these cases would strengthen the conclusion that GECKO-A + tgBoost is a better tool for predicting SOA viscosity.

Thanks for this constructive comment. While CP successfully predicted viscosity for many types of SOA, it did not work well for modeling viscosity of indoor surface films (O'Brien et al., Environ. Sci.: Processes Impacts, 23, 559, 2021). They were mostly formed by deposition of cooking aerosols, which contain substantial amounts of high molar mass unsaturated compounds such as triglycerides and their derivatives with carbonyl and ester groups. CP significantly overestimated the measured viscosity of the films, showing limitations of viscosity predictions without considering molecular structure and functional groups. We have added the following texts in introduction:

"A method was developed to predict SOA viscosity from the $T_g$ -scaled Arrhenius plot of fragility by considering Gordon-Taylor mixing rule and hygroscopic growth of SOA particles (DeRieux et al., 2018; Shiraiwa et al., 2017). The $T_g$ compositional parameterizations (CP) and the viscosity prediction method have been applied to high resolution mass spectrometry data of various types of SOA including toluene SOA (DeRieux et al., 2018), SOA generated by diesel fuels (Song et al., 2019), β-Caryophyllene SOA (Maclean et al., 2021), and SOA generated by surrogate VOC mixtures by healthy and stressed plants (Smith et al., 2021), agreeing well with viscosity measurements. However, CP substantially overestimated viscosity measurements of indoor surface films which are mostly composed of unsaturated high molar mass compounds such as triglycerides (O'Brien et al., 2021). CP does not consider molecular structure nor functionality explicitly, representing a limitation of this method."

Lines 323-327. The authors note that the trend of N:C and O:C is consistent with previous studies, which seems correct. However, they also mention that the simulated values are 15-45% lower than these measured values. A short discussion of why these values are lower is warranted.

The discrepancies are likely due to errors on modeling gas-wall partitioning and gas-particle partitioning. In addition, the difference may be caused by missing processes in the model such as reactive uptake of oxidants and particle-phase chemistry. We have included this point in the revised manuscript. Comparisons of the simulated chemical composition to the measured data of individual compounds in the gas and in the condensed phase is needed for the series of n-alkane in order to better understand and constrain the influence of gas phase chemistry, gas-wall partitioning, gas-particle transfer, and particle reactivity on the O:C and N:C ratios, which is beyond the scope of this study but should be a subject of future studies.

The mass loadings of seed particles in the chamber were ~200-400 ug/m3, but I don't see anywhere that states the mass loadings of SOA in the chamber. Is it assumed that the SOA mass loading is equal to the seed particles mass loading?

The final SOA mass concentrations were in the range of ~300 – 6000 µg m$^{-3}$ depending on precursors (Lim & Ziemann, 2009). We clarify this information in the revised manuscript.

Lines 243-245: the authors mention that particle-phase chemistry was shown to be substantial in n-alkane SOA formation for low NOx conditions. If I understood correctly, GECKO-A models gas-phase chemistry only, and not particle-phase chemistry. The authors do not further discuss the possibility of particle-phase chemistry under high NOx conditions and how this may affect their results.

Recently, Ranney et al. (2023) suggested that cyclic hemiacetals form acetal dimers in the particle phase for SOA formed from the reaction of n-hexadecane SOA and OH/NOx. While they could not directly detect dimers, their derivatization measurements indicate the presence of acetal dimers. In the revised manuscript, we have toned down our statement to mention that oligomerization chemistry is not a dominant process and then mention these latest results by Ranney et al. (2023). We also note that the impact of such particle-phase chemistry may warrant further investigations by future model development and experimental studies.

**TECHNICAL NOTES:**
Line 76: Long-chain not defined. What value of n differentiates long-chain from medium-chain?

We are not aware of a clear definition to separate medium- and long-chain alkanes, but C8-17 have been called long-chain alkanes in previous studies.

Line 92: move "to date" later in the sentence – "The GECKO-A model is one of the most comprehensive generators of gas-phase chemical schemes to date…"
Line 107: add "the" before "oxidation".
Line 152: Should be Fig. A3a, not S3a.
Line 164: Add a comma after "In this study"
Line 165: Make "prediction" plural.
Line 194: I could be wrong, but doesn't Tenax need a registered (R) symbol following it? "Tenax®".
Line 231: "lower volatility" not "volatility lower.

Line 247: Add "an" before "effective mass accommodation".
Line 247: Change "that" to "which".
Line 251: Glass transition temperature was already defined, so the symbol Tg can be used here.
Line 252: Text suggests to look at the "green line," but no figure has been mentioned for a while. Please indicate which figure/panel.
Line 340-341 – Add "that the" before "five species".
Line 366: Add "the" before SOA.
Line 431: Reword the start of the sentence. Either "IVOCs have gained.." or "IVOCS are gaining attention.."

Thanks for carefully reading the manuscript. We have revised them as suggested.

Figure 1 – The ordering of panels is not intuitive with (d) being the top right panel.
Figure 1c - The orange line is essentially not visible behind the green line. Make some note of this in the caption or text, or visually show it some other way.
Figure 2 - The text of secondary y-axis on panel (a) and the y-axis of panel (b) are too close together.

We have revised two figures as suggested.

---

## Editor Decision (ED1)

Additional minor editorial corrections:

Lines 9-10: Suggest the following changes to the abstract: "...yet the chemical composition and phase state are poorly understood and thus poorly constrained in aerosol models." The use of "its" makes it sound like there is only one alkane SOA and hardly is a bit ambiguous.

Line 18/line 33: In the abstract you define SOA as singular but use it as a plural (e.g., n-alkane SOA adopt). I think that this is fine, and common, but then in the introduction, you define it as plural. Suggest to make it consistent in both the abstract and introduction.

Line 158: Change "is" to "are", "partitioning kinetics are"

Line 178: Suggest removing "implicitly" and the end of this sentence. Not clear what it means in this context.

Line 234: Change "is" to "are", "compounds... Torr are"

Line 265: Autoxidation and auto-oxidation are used, suggest to make consistent. The first is more common.

Line 278: The link here between "further investigations" and "future model development" is awkward. Maybe "including model development and experimental studies", or "further investigations, future model development and experimental studies".

Line 292, line 340: I agree with reviewer #2 about the use of "remarkable". I suggest notable or something with that level of emphasis.

Line 310: Change "increases" to "increase"

Line 380: Also in agreement with reviewer #2, please review manuscript carefully to differentiate modeled vs. measured results. "We also found" implies measurements. I suggest "we also predicted" to avoid confusion, particularly with the introduction of experimental results in the following sentence.

Line 385: Change "are" to "were"

Line 386: Suggest to change to "These compounds were also major products in our simulations, supporting GECKO-A..."

Line 387: Differences between the model runs? Or differences between Ranney measurements and model? Both?

Line 392: Suggest "Tg values between 157-221 K"

Line 436: Suggest removing "are", "SOA yields increased…"

Line 452: Suggest "n-alkanes"

Line 454: Is there a more precise or technical way to define "lightly oxidized" ?

Line 492: I think the capabilities of the model are overstated here, given that GECKO-A is an equilibrium-based partitioning model only. As noted elsewhere in the manuscript, some potentially important gas- and particle-phase processes are not included. I agree with the ability of the model to predict the experiments in this paper, but I don't think the model is capable of comprehensively simulating the chemical evolution of SOA.

---

## Author Response (AR2)

Additional minor editorial corrections:

Lines 9-10: Suggest the following changes to the abstract: "…yet the chemical composition and phase state are poorly understood and thus poorly constrained in aerosol models." The use of "its" makes it sound like there is only one alkane SOA and hardly is a bit ambiguous.

Line 18/line 33: In the abstract you define SOA as singular but use it as a plural (e.g., nalkane SOA adopt). I think that this is fine, and common, but then in the introduction, you define it as plural. Suggest to make it consistent in both the abstract and introduction.

Line 158: Change "is" to "are", "partitioning kinetics are"

Line 178: Suggest removing "implicitly" and the end of this sentence. Not clear what it means in this context.

Line 234: Change "is" to "are", "compounds… Torr are"

Line 265: Autoxidation and auto-oxidation are used, suggest to make consistent. The first is more common.

Line 278: The link here between "further investigations" and "future model development" is awkward. Maybe "including model development and experimental studies", or "further investigations, future model development and experimental studies".

Line 292, line 340: I agree with reviewer #2 about the use of "remarkable". I suggest notable or something with that level of emphasis.

Line 310: Change "increases" to "increase"

Line 380: Also in agreement with reviewer #2, please review manuscript carefully to differentiate modeled vs. measured results. "We also found" implies measurements. I suggest "we also predicted" to avoid confusion, particularly with the introduction of experimental results in the following sentence.

Line 385: Change "are" to "were"

Line 386: Suggest to change to "These compounds were also major products in our simulations, supporting GECKO-A…"

Line 387: DiXerences between the model runs? Or diXerences between Ranney measurements and model? Both?

Line 392: Suggest "Tg values between 157-221 K"

Line 436: Suggest removing "are", "SOA yields increased…"

Line 452: Suggest "n-alkanes"

Line 454: Is there a more precise or technical way to define "lightly oxidized" ?

Line 492: I think the capabilities of the model are overstated here, given that GECKO-A is an equilibrium-based partitioning model only. As noted elsewhere in the manuscript, some potentially important gas- and particle-phase processes are not included. I agree with the ability of the model to predict the experiments in this paper, but I don't think the model is capable of comprehensively simulating the chemical evolution of SOA.

Response: Thanks very much for carefully going over the manuscript. We have implemented all suggested changes. Please see the revised manuscript with track changes.